# Robust Fabrication of Polymeric Nanowire with Anodic Aluminum Oxide Templates

**DOI:** 10.3390/mi11010046

**Published:** 2019-12-30

**Authors:** Larry Brock, Jian Sheng

**Affiliations:** Department of Engineering, Texas A&M University Corpus Christi, Corpus Christi, TX 78412, USA; larry.brock@ttu.edu

**Keywords:** nanofabrication, anodic aluminum oxide templates (AAO), polymeric nanowire

## Abstract

Functionalization of a surface with biomimetic nano-/micro-scale roughness (wires) has attracted significant interests in surface science and engineering as well as has inspired many real-world applications including anti-fouling and superhydrophobic surfaces. Although methods relying on lithography include soft-lithography greatly increase our abilities in structuring hard surfaces with engineered nano-/micro-topologies mimicking real-world counterparts, such as lotus leaves, rose petals, and gecko toe pads, scalable tools enabling us to pattern polymeric substrates with the same structures are largely absent in literature. Here we present a robust and simple technique combining anodic aluminum oxide (AAO) templating and vacuum-assisted molding to fabricate nanowires over polymeric substrates. We have demonstrated the efficacy and robustness of the technique by successfully fabricating nanowires with large aspect ratios (>25) using several common soft materials including both cross-linking polymers and thermal plastics. Furthermore, a model is also developed to determine the length and molding time based on nanowires material properties (e.g., viscosity and interfacial tension) and operational parameters (e.g., pressure, vacuum, and AAO template dimension). Applying the technique, we have further demonstrated the confinement effects on polymeric crosslinking processes and shown substantial lengthening of the curing time.

## 1. Introduction

Generating functional surfaces with micro-scale structured roughness (e.g., multi-tiered biomimetic structures [1]) have received substantial attention recently. It is found that surface mounted micro-/nano-scale pillary structures, which mimic natural counterparts such as hairs in gecko toe pads, multi-tier roughness on a lotus leaf, and hairs over a rose petal, enable us to tune the surface energy through topology of these micro-structures and their spatial arrangements. Although micro-scale structures are relatively easy to fabricate by lithography including photo- and soft-lithography, as well as other conventional Si-based etching techniques, nano-scale structures (or nanowires), are significantly difficult to create, especially when they are synthesized on the top of those micro-scale structures, especially for polymeric materials. 

In literature, several groups have successfully used anodized aluminum oxide (AAO) that under proper anodizing conditions contains very well structured nanopores as templates to fabricate nanoscale roughness [2,3,4,5,6,7,8,9]. With advances in polymer sciences and discovery of precursor wetting films [10], nanotubes and nanofibers have been successfully fabricated with several polymers [2,3,4,5,6,7,8,9,11,12,13,14,15,16,17,18,19,20,21,22,23,24,25,26,27,28,29,30] using methods often termed as “template synthesis” [31,32]. Detailed reviews can be found in [4,26,28,29,30]. Due to the methods being highly dependent on wetting properties of polymers and templates, they have often been applied to acrylate polymers such as poly(methyl methacrylate) (PMMA), poly(vinylidene fluoride) (PVDF), polyethylene oxide (PEO), polystyrene (PS), and polytetrafluoroethylene (PTFE), as well as polymers in solvents such as poly(viny alcohol) (PVA) and poly(vinyl chloride) (PVC), etc. Although demonstrated with exceptional capabilities, the methods’ dependency on specific materials (but excluding many cross-linking elastomers like polydimethylsiloxane, polyurethane, etc.) and often associated cumbersome and expensive chemical processes make them too costly to structure a smaller area (e.g., centimeters) and prevent them from scaling up to pattern a large surface (e.g., meters) in real-world applications. 

In this paper, we report a further development of the robust and scalable fabrication technique that is applicable to a wide range of polymers including elastomers, thermosets and thermoplastics to generate nanowires with high aspect ratio using vacuum-assisted molding over AAO templates. Differing from template wetting methods utilizing “spontaneous” wetting of nano-pours by precursor films [23,28,29,30] or “nonspontaneous” infiltration by pressure [26], the presented method amplifies the small scale physical mechanisms like surface tension and spreading under vacuum to drive the material into structured nanopores in the AAO template. After the polymer has infiltrated the entire AAO template, polymer nanowires will allow to cross-link and cure within nanopores. The template is then removed completely afterward by wet etching procedures with either compatible acid or alkaline, and subsequently release nanowires. The consumption of the template seems to be wasteful, but our ability to cost-effectively produce AAOs at large scales (Section 2.1, [33]) has greatly reduced the fabrication cost and enabled the proposed technique highly scalable. Anecdotally, we have fabricated a polydimethylsiloxane (PDMS) surface with PDMS nanowires of 0.75 m × 0.25 m with a scaled-up apparatus of 1 m × 0.5 m × 0.4 m (the latest being the thickness) similar to the one presented in the paper. Owing to its weak dependency on material wetting properties, it can be applied to a wide range of polymers including thermoplastics (polymer melts), solutions and most of cross-linking polymers including elastomers. To demonstrate its robustness, we have applied the method to four cross-linking polymers (PDMS; polyurethane, PU; commercial two-part epoxy; cyanoacrylate, CA) and four thermoplastics (polyvinylidene fluoride, PVDF; polyamide, PA; polytetrafluoroethylene, PTFE; and high-density polyethylene, HDPE). Results showed that high aspect ratio (75:1) nanowires are invariably fabricated by those diametrically different materials. The confinement effects on cross-linking are also investigated and found that confinement has substantially lengthened the process. Differing from those template wetting methods in the literature, the proposed method, however, is applicable to any cross-linking polymers including elastomers and thermoplastics. Note that no prior literatures have reported successful synthesis of elastomer nanowires using AAO templates, nor the precursor film wetting under the vacuum conditions. It is also worth to note that all polymers used in our study are commercially available and not prepared with any costly mechanical and chemical processes prior to experiments or any expensive equipment (e.g., high pressure heated press). The only equipment needed is a low-cost vacuum chamber (e.g., fabricated from aluminum blocks). Compounded with cost-effective fabrication of large AAO templates, the proposed method is highly scalable to address the needs of real-world applications such as anti-fouling (AF) or foul-releasing (FR) ship hull coating. In the following, we first summarize the fabrication method of the AAO template over a large substrate (Section 2.1), followed by detailed descriptions of vacuum-assisted molding for both cross-linking polymer and thermoplastic (Section 2.2) as well as on nanowire releasing procedures (Section 2.3). In discussion (Section 3), a model describing infiltration length and time for the molding process are developed and followed by SEM micrographs of nanowires. The implications of confinement on fabrication, especially cross-linking, are discussed in Section 3.3.

## 2. Materials and Methods

### 2.1. Fabrication of Anodized Aluminum Oxide (AAO) Template over a Large Substrate

Only recently, the self-organized nano-porous in the anodized aluminum oxide (AAO) [34] under proper conditions are discovered and rapidly found its way in applications of generating nano-sensors or actuators using AAO as negative template [35]. In the following, we will first briefly summarize key formation mechanisms of these nanopores in the AAO before discussing our protocol used in this work to fabricate AAO templates over a large area (e.g. 75 mm × 75 mm). 

AAO is formed by an electrochemical process occurring at both metal-oxide and oxide-electrolyte interfaces [36]. At the beginning of the anodization, a barrier layer is formed as soon as the current is applied (e.g., a few seconds). As the anodization continues, small fissures formed at the outer surface of the barrier layer propagate into the aluminum substrate and expand in the lateral directions to form pore-like structures [37]. Eventually, energy density within the aluminum substrate achieves equilibrium that causes pore structures to be self-organized as a closely packed array of hexagonal cells, each containing a cylindrical central pore that is perpendicular to the aluminum substrate ([36] and Figure 1). The thin non-porous barrier layer has a hemispherical and scalloped geometry that is half of the pore wall thickness [36]. The size and distribution of these scallops are affected strongly by anodization conditions including voltage, pH level, and temperature. At the scalloped portion of the aluminum/oxide interface, the pore bases are not homogeneous both in-depth and in pore distribution where the oxide layer contains a charge accumulated during the growth that reaches a maximum when the fissure appears in the outer oxide layer, as it incorporates impurities into the oxide layer [38]. While the growth of the oxide constantly opposes the dissolution of oxide, the inward movement of oxide into the substrate regulates pore nucleation and arrangement by the applied electric field [37].

To form regularly spaced nanostructures in an AAO layer, anodization must involve the following steps: (i) initial anodization (Figure 1a) to establish electoral density distribution over an aluminum substrate, (ii) remove the first oxide layer to establish an organized barrier layer (Figure 1b), which is crucial to create highly ordered nano-pores. (iii) Secondary anodizing to allow pores to be developed into the substrate (Figure 1c). It is found that time and temperature strongly affect the depth of pores [39,40], while anodizing voltage controls spacing and the pH level of the anodizing electrolyte affects pore diameter. Interplaying of these parameters (temperature, PH, and voltage) provides us with a large parameter space to generate a wide variety of nanopores (Figure 2). For instance, long (>30 µm) straight pores with a thin-wall (Figure 2a) can be achieved by anodizing for long periods of time (4 h) at a low temperature (e.g., 1 °C) and a low voltage (e.g., 60 V), namely soft anodizing. Note that maintaining low electrolyte temperature is crucial to generate a very thick AAO layer since it prevents thermal damages to these nanostructures. When anodized at higher voltage (e.g., 140 V) and lower temperature (−3 °C) for a short period of time (e.g., 2 min), namely hard anodizing, a thick-wall small-pore AAO layer can be fabricated (Figure 2b). It is noteworthy that low temperature (<0 °C) reduces the transport of Al^3+^ and consequently causes thicker wall and smaller pore size. Combinatorial applications of these processes (e.g., soft and hard anodizing) in sequences and varying anodization parameters (e.g., voltage, temperature, and pH level) allow us to create much complex nanopore structures. Figure 2c demonstrates that a tiered nanopore, i.e., large-diameter thin-wall pore on the top of several small diameter thick wall pores are produced by a soft anodizing at 40 V followed by a hard anodizing at 130 V. 

In this work, all AAO templates (75 mm × 75 mm) are produced by an in-house developed device that consists of a 500 mL container with a cover containing two carbon electrodes separated at a distance of 100 mm, a cooling unit that circulates anodizing electrolytes by a 1.5 hp chemical resistant pump through a stainless steel heat exchanger (1.2 m × 0.8 m) submerged in a liquid nitrogen (LN) bath. The device is cheap to construct and operate, capable of performing hard anodizing of a large sample (75 mm × 75 mm) at 130V and −3 °C for an hour without refilling LN bath. The AAO sample preparation is started that an aluminum piece (e.g., 75 mm × 75 mm × 0.25 mm, the latest being the thickness) is cut off from a large annealed aluminum coil (300 mm wide and 4 m long, 99.99% purity, Goodfellow Cambridge Limited, Huntingdon, UK). The sample is first flattened by a roller and mechanically polished. After mechanical polishing, the sample is further processed by an electro-polishing process in the phosphoric acid (14.8 M H3PO4, Sigma–Aldrich, St. Louis, MO, USA) at 55 °C for 12 min. The power supplied is set at a constant current mode with the current of 3 A and initial voltage of 40 V. As the process progresses, the voltage will slowly reduce to 9 V. To achieve optimal results, the polishing is performed in a 500 mL beaker on a magnetic hot plat with a stirrer to remove micro-bubbles formed at the sample surface, which is attached to the carbon anode with a nylon screw. After polishing, the sample must be thoroughly rinsed with DI water (1 MΩ, Millipore, Sigma-Aldrich, St. Louis, MO, USA) immediately. First anodizing is then carried out in oxalic acid (0.3M C_2_H_2_O_4_, Sigma–Aldrich) with power supply in constant voltage mode. Varying anodizing voltage results in different pore diameters and interpore distance (e.g., 40 V produces 55 nm diameter pore, 60 V generate 75 nm pore, and 80 V 125 nm pore). Note that in practice, the anodizing at low voltages (<120 V) is often referred as “soft” anodizing, while those at high voltages (120–150 V) as “hard” anodizing. After 12 min, the self-organized structures and barrier layers are formed. This initially formed oxide layer is then etched away in a mixture of chromic (H_2_CrO_4_, 1.6% v/v) and phosphoric acid (H_3_PO_4_, 6% v/v) at 60 °C for 16 min to produce a well-defined and organized pattern. The sample is then rinsed with DI water before the second anodization in oxalic acid (0.3 M C_2_H_2_O_4_, Sigma–Aldrich). The voltage of the second anodizing must be set at the same as the first anodizing initially, but can be varied during the process to produce the tiered structure (e.g., Figure 2c). While the length of the processing time determines the pore depth, but the rate is greatly affected by temperature. For instance, at 10 °C the anodizing rate reaches 55 nm/min, while at 0 °C the rate is ~200 nm/min. Figure 2b shows that a 6 µm deep pore is produced by a 60 min anodizing at 0 °C. For those AAO templates used in the following studies, soft anodizing at 7 °C and 60 V are performed for 12 min resulting in a pore depth of 2 µm. 

### 2.2. Vacuum-Assisted Molding

Once an AAO template is made, nanowires can be fabricated by filling nano-pores with desired material such as metal [41,42,43,44,45,46,47], metal oxide [44,48,49,50,51,52,53] and polymeric materials [2,3,4,5,6,7,8,9,11,12,13]. To deposit metal and metal oxide, methods such as sputtering [54,55,56,57,58], evaporation [46,55,56,57,59,60,61] and thermo-embossing of metallic glass [62,63,64], are widely used to fill nanopores and form nanowires. The nanowires are then released by either wet or dry etching of the templates. With these methods, the high aspect ratio and fully erected metal/metal-oxide nanowires have been fabricated. 

However, In the case of nanowires made of polymers or plastics, molding and thermos-embossing of these materials and then releasing them by either etching or peeling from AAO template are often the most cost-effective methods [65,66,67]. To effectively replicate polymeric nanowires within AAO templates, the polymers must first infiltrate nano-pores completely. By exploring precursor films of low surface energy liquids in wetting high surface energy solids, several groups have successfully fabricated nanotubes and nanowires with polymer melts and solutions [4,23,26,27,28,29,30]. Although these wetting based techniques are highly successfully, the infiltration processes (e.g., infiltration time, depth and wire geometry) rely on maintaining the *local* disparity between the lower surface energy of polymer liquids and the higher surface energy of AAO templates (i.e., precursor wetting film is a microscopic mechanism) [10], which inevitably results in highly inhomogeneous nanowires over a large AAO template since variations in polymer liquids and nanopores in templates are unavoidable. Moreover, the strict requirement of surficial energy imbalance limits methods to polymers with high mobilities (e.g., PEO, PVDF, PMMA, PS, PTFE, etc.) or polymers in solvent. For many thermoset elastomers (e.g., PDMS, polyurethane, epoxy, etc.) often with high surface energy, wetting-based methods appear to be ineffective. To our best knowledge, there are no reports on successful applications of wetting-based methods to elastomers. On the other hand, vacuum-based methods [26] relying on external force induced infiltration are expected to be applicable to both polymer melts and thermoset elastomers, which inspired current research. Note at micron or millimeter scales inertia forces such as pressure and gravity provide effective means, i.e., the soft polymers are cast onto the AAO template and placed under vacuum, under which the air trapped within the microscale pores expands into bubbles and subsequently removed by buoyancy, meanwhile the materials flow into these pores. As the pore size reduces down to nanometers, those inertial forces play less and less important roles in comparison to the interfacial forces (e.g., interfacial tensions and capillary forces), which renders the infiltration of nanopores by inertia (e.g., pressure and gravity) ineffective. Baek et al. [67] have fabricated RR-P3HT (Regioregular poly(3-hexylthiophene)) nano-structures using capillary force to drive RR-P3HT into the nano-pore completely. With an argument of the balance between capillary force and weight, they provide the infiltration depth scaling with surface tension proportionally and pore diameter inversely. They have reported a maximum infiltration depth of ~1um and speculated that the air trapped within pores prevents the filling of the entire pore.

In this paper, we argue that since at nano-scale, interfacial mechanisms like capillarity are dominant in comparison to inertia, a universal method that amplifies these mechanisms can be developed for polymers to infiltrate the entire pore depth. Detailed analysis (Section 3.1) shows that the process is driven by interfacial forces including capillary forces and counteracted by the pressure difference between the trapped air in the pore and the ambient as well as by gravity. Own to the large surface-to-volume ratio at nano-scales, the full infiltration can be easily achieved by maximizing the balance between surface tension and pressure difference through vacuum. Hence, we propose a simple in-situ vacuum-assisted molding procedure that includes (i) evacuating the air from the template and polymers concurrently; (ii) casting soft material onto the template under vacuum and allow it to fill pore completely; and (iii) curing or solidification of materials under controlled ambient pressure. Our method differs from conventional molding methods in the fact that we remove the air from pore first such that we can minimize the pressure difference between pore and ambient. It can be shown later that neither density nor viscosity of the material is crucial parameters in controlling filling efficacy. Consequently, the method is applicable to a wide variety of soft materials. To demonstrate its versatility, two categories of soft materials (polymers and thermoplastics) are used in the current paper. Figure 3 shows the schematics of the vacuum-assisted molding procedure. To provide better control over chamber pressure, we have constructed a 20 cm × 20 cm × 40 cm vacuum molding chamber that consists of two separate half chambers sealed with screws and gaskets. This unique design allows us to control and vary the chamber pressure precisely during the molding process. Since both chambers are machined from two solid aluminum blocks with a minimum wall thickness of 4 cm, the chamber can operate between ~0.1 Pa and 1 MPa. The largest processing surface is 12 cm × 12 cm at one time. Shown in the Figure 3a, the chamber, consisting of two half chambers: one for holding polymer material and the other AAO template, allows us to perform in-situ degassing of polymer and evacuating air from template concurrently, as well as to provide external heat to cure the polymer or melt thermoplastics under vacuum.

The molding procedure was the following: for both types of materials, AAO template was attached securely at the top wall of the chamber (Figure 3a) with adhesive. For polymers such as PDMS, polymethyl methacrylate (PMMA), polyurethane and epoxy etc., after thorough mixing, it was placed in the bottom half of the chamber first. The chamber was then assembled and completely evacuated of air, which degased the polymer mixture and AAO template concurrently. After high-vacuum (~2.7 Pa or 20 mTorr) was maintained for a sufficiently long time, the chamber is flipped which allowed the polymer to pour over the template and subsequently to infiltrate pores with capillary forces as only effective mechanism. When the polymer uniformly covered the entire template, the chamber was pressurized and heated for curing. (Figure 3b). Note that the template and polymer were only vacuumed for long enough to only allow gas to be removed. Degassing for excessively long time can result in the removal of the solvent from the mixture and formation of many microbubbles (1~10 μm) in the polymer matrix. Additionally, the polymer is cured within a pressurized chamber instead of under vacuum. This subtle deviation from the conventional molding technique was found to be necessary. We have observed that under vacuum most solvents used in two parts polymer exceeds their vapor pressure and results in preferential degassing within the pore. This preferential degas results in not only shortened nanowire but also substantial changes and homogeneity of material properties of the fabricated nanowires. While molding polymer nanowire from AAO template involves precise timing and many steps, synthesis of thermoplastic nanowires is considerably simpler: the thermoplastics is placed on top of the AAO template before the chamber is assembled and vacuumed. The entire chamber is then heated by a large hotplate with a heating surface of 0.4 m×0.4 m to temperatures above the melting temperatures ($T_{m}$) of materials or at least to glassy transition temperatures ($T_{g}$) (Table A1), which allows material to enter the pore. Due to plastics used all being commercially available sheets or blocks with unpolished surfaces, we used temperatures slightly above $T_{m}$ in all our experiments. Once the infiltration is complete, it is solidified within the pore and nanowire formed. Solidification can be performed either under vacuum or with pressure, although solidification under vacuum does prevent oxidation of the material (Figure 3c). To improve the contact between plastics and template, a small weight (e.g., 100 g) was also applied. We need to emphasize that the capillary force was the dominant mechanism and the application of a weight only enhanced it. This feature of the proposed method differs substantially from those thermo-embossing methods that the application of large pressure is often required [68]. 

To demonstrate the range of the applications of the proposed method, we have fabricated nanowires with assorted materials. The cross-linking polymers used in the following experiments are polydimethylsiloxane (Sylgard 184, Dow, Houston, TX, USA), polyurethane (TC-9445, Burman Industries, Van Nuys, CA, USA), polyurethane rubber (V60, Polytek, Easton, PA, USA), commercial two-part epoxy (E60NC, Henkel Adhesive Technology, Rocky Hill, CT, USA), and cyanoacrylate (commonly known as “crazy” glue). All cross-linking polymers are commercial products as kits of two-part solutions. The thermoplastics are polyamide (Nylon, DuPont, Wilmington, DE, USA), PVDF (Quantum plastics inc., Elgin, IL, USA), PTFE (McMaster-Carr, Elmhurst, IL, USA), and high-density polyethylene (HDPE, McMaster-Carr). All thermoplastics were used in the form of either a 9 mm thick sheet or a 25 mm thick blocks. Note, all polymers were used in their commercially available package (i.e., no additional treatments have been applied). Additional, after anodizing, the AAO templates were washed with acetone, methanol, isopropanol in sequence, rinsed with DI water (Millipore, 10 MΩ), and finally vacuum dried at 80 °C and 2.7 Pa for 48 h. No additional treatments have been applied to the templates either. The materials and their corresponding processing parameters are summarized in Table 1.

### 2.3. Release Nanowires

Once polymer nanowires are molded, the AAO template including its aluminum substrate needs to be removed by acids or base solutions. Since most methods have the potential to cause damages to nano-structures, the release method must be carefully selected to remove both aluminum substrate and AAO effectively while minimizing damages to polymeric nanowires. We have found that for cross-linking polymers such as PDMS and polyurethane, base solutions at low molar concentration and low temperature are preferred; while for thermoplastics such as PVDF, PTFE, and high-density polyethylene (HDPE) both acids and base solutions at low concentrations can be used. In the current paper, sodium hydroxide (NaOH, 1M) is used for polymers and hydrochloric acid (HCl, 5M) is used for curable epoxy and all thermoplastics. The wet etching is performed at a constant temperature of 5 °C under constant stirring. Etching parameters are summarized in Table 1.

## 3. Results

### 3.1. Effective Infiltration of Nano-Pores with Vacuum Assisted Molding

To better understand the process and control space for precise fabrication, we modeled the filling process under vacuum with lubrication theory to provide a better guideline. It is worth to point out that neglecting precursor film in our model can be justified since under high vacuum (~2.7 Pa) the surface energy of a liquid increases greatly while that of a solid remains the same or increases only slightly, consequently reduces the energy imbalance. This reduction in energy imbalance between liquids and template would arguably reduce the extent of precursor thin film especially in case of elastomers. Although being an interesting future research topic, it is certainly beyond the scope of the current paper. Here, we consider the material infiltrates a single pore with the depth of H and the radius of R as a continuum slug moving in a close-ended cylinder (Figure 3d), the equation of motion of the slug is:(1)$$\underset{\mathrm{Inertia}}{\underbrace{m_{p}\frac{dU_{p}}{dt}}}=\underset{\mathrm{Capillary}}{\underbrace{2\pi R\gamma \cos\theta }}+\underset{\mathrm{Pressure}}{\underbrace{\pi R^{2}\left(p_{0}-p_{a}\right)}}+\underset{\mathrm{Gravity}}{\underbrace{m_{p}g}}-\underset{\mathrm{Drag}}{\underbrace{2\pi Rz\sigma_{r}}},$$
where $m_{p}=\pi R^{2}z$ is the mass of polymer slug within the pore, z is the infiltration depth, and $U_{p}$ is the infiltration speed. Equation 1 shows the balance between body force (i.e., gravity), and surface forces including capillary, pressure forces and viscous drag. At nano-scales, the inertia (left-hand side term in Equation (1)) is negligible, which results in the instantaneous balance of all forces. The first term denotes the capillary force where γ is the surface tension of polymer material over a smooth alumina substrate and θ is its equilibrium contact angle. The second term represents the force due to the pressure gradient, where $p_{0}$ and $p_{a}$ are the pressure of the ambient and air trapped in the pore, respectively. For simplicity, $p_{0}$ is assumed to be constant and $p_{a}$ increases as the slug penetrates the pore that is approximated as the ideal gas under the isentropic compression, $p_{a}\left(z\right)=p_{a}^{0}{\left(\frac{H}{H-z}\right)}^{\kappa }$, where κ is the ratio of specific heats and $p_{a}^{0}$ is the initial pressure of air trapped within the pore. This isentropic assumption may introduce errors in modeling $p_{a}$, but the deviations are expected insignificant in practice. The third term, gravity force, is depth-dependent and can be approximated as $\rho \pi R^{2}zg$, while the last term represents the viscous drag as the material fills the pore. With the assumption of the entire slug moving at the same speed as the front, the viscous drag is approximated as $\sigma_{r}=\mu \frac{dz}{dt}\cdot \frac{2}{R}$, where μ is the dynamic viscosity and $\sigma_{r}$ is the skin friction. The equation of motion can be simplified to the following: $$\frac{dz}{dt}=\frac{R\gamma \cos\theta }{2\mu z}+\frac{R^{2}p_{0}}{4\mu z}\left[1-\frac{p_{a}^{0}}{p_{0}}{\left(\frac{H}{H-z}\right)}^{\kappa }\right]+\frac{\rho gR^{2}}{4\mu }$$

To better understand, we represent it in its dimensionless form: (2)$$\frac{dz^{*}}{dt^{*}}=\frac{\cos\theta }{2Ca}\left(\frac{R^{*}}{z^{*}}\right)+\frac{Re}{8}\left[1-\frac{p_{a}^{0}}{p_{0}}{\left(\frac{H}{H-z}\right)}^{\kappa }\right]p_{0}^{*}\left(\frac{R^{*}{}^{2}}{z^{*}}\right)+\frac{Bo}{4}R^{*}{}^{2},$$
where “*” denotes the quantity normalized by characteristic scales. The characteristic length scale is the pore depth, H, the temporal scale is the 1D mass diffusion time of material defined as $T=H^{2}/2D$, where D is the mass diffusivity and the diffusion velocity, U=H/T=2D/H. Note that for most polymers, mass diffusivity is $D\ll 10^{-9}\;m^{2}/s$ in solvents or almost zero in air, and the diffusion speed is $U\ll 10^{-3}m/s$ in solvents and almost zero in air. Three dimensionless numbers are obtained as capillary number (Ca=μU/γ), Renolds number (Re=UH/ν), and Bond number ($Bo=gH^{2}/\nu$), which suggests three key mechanisms affecting the infiltration process. The *Ca* (the first term) represents the effect of viscous force to that of surface tension, which is expected to be Ca≪1 and consequently the largest contribution to the change of z. Note that the surface property represented by equilibrium contact angle affects the direction of the material propagation, i.e., amphiphobic materials (cosθ<0) will not fill the pore, while amphiphilic materials (cosθ>0) will. To exam the effect of pressure-driven inertia represented by *Re* (i.e., the ratio between inertia and viscous forces), for most polymer like PDMS or thermoplastics, *Re* is ≪1. This observation suggests that pressure force contributes little to fill the nanopore. Furthermore, the sign of the second term in Equation (2) can be controlled by adjusting the ratio of the initial pore pressure, $p_{a}^{0}$, and $p_{0}$. In the case of equal pressure conditions, the contribution is always negative. The last term in Equation (2) signifies the importance of buoyancy or gravity. For instance, to fill a 1-um deep pore with PDMS (μ ~3500 cP), the Bo number is $\sim O\left(10^{-10}\right)$. 

Briefly, the above analysis supports the assertion that in nanoscale, the capillary force is the most effective driven mechanism to fill the pore, while the inertia forces, pressure, and buoyancy, are ineffective means. The filling into and repelling from the pore can be effectively manipulated by altering the surface property, but they can be medicated by adjusting the pressure difference between ambient and pore cavity. With substantial algebra, the final length of nanowire and the time to reach that length can be obtained analytically as:(3)$$\frac{z_{f}}{H}=1-{\left[\frac{p_{0}}{p_{a}^{0}}\left(1+C_{f}\right)\right]}^{-1/k},$$
(4)$$t_{f}=2\left(\frac{\mu }{\gamma \cos\theta /R}\right){\left(\frac{H}{R}\right)}^{2}\int_{0}^{\frac{z_{f}}{H}}\frac{zdz}{\left\{1+\frac{1}{2C_{f}}\left[1-\left(\frac{p_{a}^{0}}{p_{0}}\right){\left(1-z\right)}^{-k}\right]\right\}}\;,$$
where $C_{f}$ is a non-dimensional quantity, $C_{f}=\gamma \cos\left(\theta \right)/p_{0}R$ which compares the surface stress, γcosθ/R to the counteracting pressure, $p_{0}$. Notice that the constant is, in fact, the combination of *Re* based on pore radius ($Re_{R}=p_{a}^{0}R/\mu U$) and *Ca*-based on pore radius ($Ca_{R}=\mu U/\gamma$). Additionally, one further notice that the viscosity is completely absent in $C_{f}$ and consequently, be absent in the expression of $z_{f}$, but the time to fill the pore-scale linearly with it. The coefficient, $\frac{\mu }{\gamma \cos\theta /R}$, in Equation (4) bearing the unit of time, can be interpreted as the characteristic time scale as the viscous fluids being driven by interfacial forces. 

We summarize these observations graphically in Figure 4. The normalized final length, $z_{f}/H$ and the normalized filling time, $t_{f}/\left(\mu /p_{a}^{0}\right)$, are described in Figure 4a,b. Note that $t_{f}$ is only normalized by $\mu /p_{a}^{0}$, which suggests that only viscosity and inertia of the material intrinsically determine how long it takes to complete the infiltration. Figure 4a,b shows that as surface tension increases (i.e., $C_{f}\gg \;1$), the material will completely infiltrate the pore at a short time; whereas the surface tension is smaller than the pressure resistance (i.e., $C_{f}<\;1$), the pore can only be partially filled and it will take an almost infinite amount of time (Figure 4b). It is also shown that the time to fill the pore is scaled with the square of the aspect ratio, H/R. The former scenario corresponds to our vacuum-assisted molding of nano-pores, and the latter corresponds to the molding of pores with a much larger radius (e.g., R≫10 μm). To further provide experimental guideline for different material over various AAO templates, we have provided wire parameters, $z_{f}$ and $t_{f}$, for molding PDMS (Figure 4c) and PTFE (Figure 4d). These parameters are computed with respect to pore radius, R, and initial pore pressure, $p_{a}^{0}$. The $z_{f}/H$ are shown as the labeled contour lines superimposed over the grayscale contour plot of the infiltration time, $t_{f}$. The aspect ratio used in Figure 4c,d is unity (i.e., H/R=1). The processing parameters of both polymer PDMS and thermoplastic PTFE show similar distributions, i.e., as the pore radius increases the length of wire reduces; so does it as $p_{a}^{0}$ increases. It can be shown that due to the significant reduction in the surface tension of PTFE at molding temperature, 320 °C, the processing parameters are much more sensitive in comparison to those in producing PDMS nanowire. It is clearly that under current experimental conditions, i.e., R<1 μm and $p_{a}^{0}=$2.6 Pa, full-length nanowire can be produced almost instantaneously as soon as the materials make the contact with the template (<1.2 s). These analyses further reveal that the proposed vacuum-assisted molding can be applied to a wide range of polymeric and thermoplastic materials. The processing regime indicated in contour of $\left(R,\;p_{a}^{0}\right)$ plane allowing successful molding is very large. This prediction agrees very well with the method’s applicability in a wide range of polymers used in the later experiments (i.e., from cross-linking elastomer, thermosets to various polymer melts). Note the proposed model is aimed to provide a better mechanistic understanding of vacuum-assisted pore filtration and provide qualitative guidelines to the scaled-up fabrication procedures. Detailed validation of this model by quantifying the infiltration depth and infiltration time vs. initial pressure and pore diameter requires further experimentation and will be communicated in the future.

### 3.2. Replications of Nanowires

To demonstrate the robustness of our fabrication method, we have applied it to several characteristic polymeric and thermoplastic materials. The nanowires were molded over the templates with the same specifications: pore diameter is 75 nm and pore depth is 2 μm. The AAO templates were produced by a two-step anodization process over an aluminum sheet with a nominal thickness of 800 μm. Each template was initially cut from a large sheet of coil to form a small piece of $50\times 60\;{\mathrm{mm}}^{2}$, and mechanically polished by a roller press. The aluminum coil piece is electro-polished in the phosphoric acid solution (14.8M H_3_PO_4_) at 55 °C for 12 min. The sample is then first anodized at 60 V in oxalic acid (0.3M C_2_H_2_O_4_) at 7 °C for 12 min. After the first anodized alumina film is stripped away at 55 °C for 16 min in a mixture solution of phosphoric acid (6% w/w) and chromic acid (5% w/w), the second anodization is performed in 0.3M C_2_H_2_O_4_ at 7 °C for 12 min. A highly ordered AAO template is formed.

The vacuum-assisted molding of various polymers and thermoplastics were performed. The polymers used in this experiment are PDMS, Polyurethane 9445, Epoxy and superglue (cyanoacrylate), while thermoplastics include PVDF, PTFE, polyamide, and HDPE. The fabrication parameters are summarized in Table 1. The materials are specifically selected to showcase the capability of technique and their potential applications, as well as to demonstrate the wide varieties of characteristics of surfaces covered with nanowires. For instance, PDMS is a widely used material for microfluidics and microelectromechanical systems (MEMS) devices owing to its biocompatibility and easiness in device fabrication. Due to its relatively high viscosity (μ=3500 mPa·s) and low tearing stress, it is often considered a difficult material to create and release the nanowire using AAO template-assisted synthesis methods. As discussed in Section 3.1 analytically and demonstrated experimentally in Figure 5a, the high viscosity of PDMS does not prevent creating high aspect ratio nanowire when vacuum-assisted molding technique is used. This result supports our assertion that the viscosity of the material is not a control parameter in the fabrication of nanowires. However, fabricating PDMS nanowire is not a simple task that is often complicated by the complexity in material properties, such as stiffness and tearing stress etc., and most importantly, variable polymerization under confinement that will be further discussed in the following section. It is observed that these complexities cause wide-spread topologies due to the bundling of nanowires after the release from the template. In Figure 5a, we showcased one of many PDMS nano-wire surfaces but the most striking pattern, which in sub-micron scales the surface roughness by bundling of nano-wires resembles the surface of a cauliflower in macro-scale. Further surface characterization by water contact angles (WCA) over these surfaces patterned with polymer nanowires is performed by an in-house developed goniometer. All WCA experiments hereinafter are conducted with a 5 μL DI water (Millipore, > 10 MΩ) sessile drop, imaged by a 2 K × 2 K CCD camera (Imperx IPX-4M15L) and analyzed by ImageJ. The result reveals that this “cauliflower-like” nano-roughness improve the water contact angle substantially to 142°±3° from 110°±3°. Although fascinating, we have not conducted any systematic study on how to make surfaces with well-controlled nano-roughness patterns, neither have we identified the key mechanisms in this process. Due to its popularity in the biomimetic adhesive and contact mechanics community, such as glueless gecko adhesive [69], we have created a nanowire surface using Polyurethane. Two polyurethane, Volvo V60 and 9445, were used but only result from 9445 was shown in Figure 5b. Owing to its high stiffness and relatively low surface energy (water contact angle of 88°±3°) by Polyurethane 9445, straightly erected nano-pillars were successfully fabricated. To demonstrate the method’s versatility, we also applied the method to two commercially available materials: two-part epoxy (E60ENC) and superglue (cyanoacrylate). Both materials produced surprising surface patterns, e.g., lotus leaf-like roughness patterns were clearly observed on epoxy nanowire surface (Figure 5c), and complicated but periodic nano-patterns were also observed on super glue surface (Figure 5d). Since both materials have relatively high Young’s modulus and shorter fabrication time (Table 1), the individual nanowire can be clearly identified in the SEM micrograph. 

To further demonstrate the advantage of our proposed method which amplifies key mechanisms most suitable in nanoscale fabrication, we applied the method to thermoplastic materials with minimum modification. Figure 5e–h show results using four commonly used thermoplastics: PVDF, Polyamide, PTFE, and HDPE. In Figure 5e, we have demonstrated that PVDF nanowire with a high aspect ratio can be easily produced. Although it is a stiff material, the wires were entangled and bundled at the tip that resulted in an increased porosity. The dependency of nano-roughness topology on material properties can be shown clearly by “nanograss” formed by Polyamide (Figure 5f). High strength and low surface energy allowed the formation of straight individual rigid nanowire. Owing to their resistance to a wide range of chemicals, we have tested our method to create PTFE and HDPE nanowires. The substantial modification of wetting properties of the surface by those surface mounted nanowires in these two materials have been clearly observed. For instance, the water contact angle of the PTFE surface with nanowire improve to 142°±3°. Note that the nanowire made of thermoplastic materials have uniform length over the entire sample surface area of $50\times 60\;{\mathrm{mm}}^{2}$. This uniformity in nanostructure over a large area is not easily achieved by other existing methods, such as thermos embossing. The relative ease of our proposed method in generating nanowire over a large surface area is not a surprise since our method amplifies one of the dominant transport mechanisms in nano-scales.

### 3.3. Implications of Polymerization under Confinement

Although the proposed method is robust and allows fabrication of high aspect ratio nanowires by AAO templates, we have observed early on that the curing time of polymer nanowire is substantially longer than that in bulk, which results in substantial variations in the surface topology of released nanowires if they are not properly cured. To investigate this phenomenon further for generating a more robust fabrication protocol, we have first conducted an experiment on determining the curing time of co-polymer PDMS under confinements.

In this experiment, the templates were created using the same anodization procedure described above, which results in nanopores of the diameter of 75 nm and the depth of 2 μm. Four templates were created and the vacuum molding was performed at 2.7 Pa (or 20 mTorr) to ensure the complete infiltration of PDMS into the nanopore. The samples were then placed in an oven at atmospheric pressure and allowed to cure at 60 °C for up to 8 weeks. One sample was removed from the oven every two weeks and the AAO template was etched by sodium hydroxide (1M NaOH) at 5 °C for two days or until all alumina and aluminum were removed. Notice that the etching was performed at low temperatures to minimize thermal damages to nanowires. We have reached to this etching protocol after repeated failures to release well-defined nanowires from the templates at higher temperatures. The nanowire surfaces were characterized by SEM. 

Results are shown in Figure 6, where those in the left column show micro-scale topology of surface owing to the bundling of PDMS nanowires cured at two, four, six and eight weeks, respectively, and those in the right are the corresponding magnified SEM micrograph to show the morphological characteristics of individual nanowire and bundling configurations. After two weeks of curing, it appears that PDMS has yet been completely crosslinked and resulted in wires that are fused together (Inset to Figure 6a) during wet etching of the template. It is reasonable to argue that PDMS near the wall of pore cures slower than those in the center that results in well define nanowire core coated with non-crosslinked PDMS. This sticky surface will bundle nearby nanowires into a microscale roughness resembling those on lotus leaf. After four weeks, more PDMS within the pore has been crosslinked, which is evident as well-defined individual nanowire although evidence of fusing among wires is still clearly observed (Figure 6b). The inset shows that there is still some cross-linking occurring that causes wires to bundle and form clusters (Figure 6c). As time progresses, the bundling reduces and individual definition of nanowire improves substantially; while after eight weeks of curing, the PDMS are completely cured and form forest-like nano-wire mat. Shown in Fig. 6d, the clear definition of individual wire is evident. Note that bulk PDMS will cure at 60 °C overnight (approximately 12 h) to form a large-scale matrix and continue to crosslink within the matrix up to 15 days. Confinement clearly lengthens these processes significantly. In the context of the present work, to generate well-defined nanowire using cross-linking polymers require significant longer curing time before they can be released from the template. 

To explore further what could be the mechanisms leading to the lengthened polymerization in confinement and subtle interplay between polymer conformation and nanopore geometry, we have conducted preliminary experiments to elucidate the key players: It is speculated that the cross-linking under nano-confinement may be hampered by several plausible mechanisms, among which long-chain polymer conformation within the pore may play a crucial role. To examine this hypothesis, we have selected two materials with distinctly different length: long-chain PDMS with loosely entangled siloxane linkage (400 repeating monomers and 100 nm nanodomain) and short length Epoxy chains with highly crosslinked hydroxide groups tightly packed (e.g., 25 repeating monomers within a 20 nm nano-domain). The templates have been created with the same two-step anodization at 60 V and 120 V, resulting in a pore diameter of 75 nm and 150 nm respectively. The second anodization time is so varied to create the appropriate pore depth of 1 μm and 2 μm respectively to maintain the aspect ratio of 13. The wires are vacuum molded and cured at 60 °C for two days, before wet etched using 1M NaOH at 5 °C. The samples are washed first and air-dried. The surface with nanowires is examined with SEM and their topology of nanowires is used to elucidate the cross-linking status. The results are shown in Figure 7. It demonstrates that at a small pore, the long-chain PDMS is not properly cured after two days (Figure 7a); whilst on the contrary, at a larger pore, PDMS is clearly cured and forms well-defined nanowire (Figure 7c). On the other hand, owing to its small molecular size, the delay in cross-linking of Epoxy in confinement has not been observed (Figure 7b,d). Note that although the aspect ratio remains the same, the microscale surface topology is drastically different and result in substantial differences in the water contact angle. For instance, PDMS surfaces (shown in Figure 7a,c) show improvement to 130±3° from 110±3° but has no clear difference between these two surface; whereas Epoxy surfaces (in Figure 7b,d) shows clear characteristics of reentrant surfaces, i.e., the water contact angles of both surfaces improve to 130±3° and 120±3° respectively from a hydrophilic material (i.e., water contact angle of Epoxy is measured at 55±3°). The difference of wetting between these two Epoxy surfaces can be attributed to the configuration of the bundling nanowires.

## 4. Discussion and Conclusions

In this paper, we present a further development of the AAO template-assisted synthesis technique that allows us to fabricate the long aspect ratio nanowires with both polymeric and thermoplastic materials. Similar to other template-assisted synthesis methods, the deposition of materials is achieved by “flowing” them at their liquid and gaseous phases into the template pores. Differing considerably from the existing thermo-embossing or molding methods that require large pressure to force the materials into nano-scale pores, the proposed method uses vacuum-assisted injection that allows materials to “flow” by itself using capillary forces without imposing excessive external pressure or utilizing body forces (e.g., gravity). Note that interfacial stresses are dominant forces intrinsic to nanoscale flows. Using a 1D creeping flow model (Equation (1)), we have derived that fabrication parameters (e.g., the length of nanowire to AAO pore depth and the time needed to reach its final length, Equations (3) and (4) can be controlled by material rheology (*Ca*), initial pore ($p_{a}^{0}$), and ambient pressure ($p_{0}$). Regime diagrams for polymers and thermoplastics (Figure 4) show a wide range of fabrication operations (e.g., for most of cross-linking polymers and thermoplastics, the filling time and aspect ratio of the nanowire is <<1 min and is >10, respectively). This unique advantage in infiltrating nano-pore with highly viscous materials allows the application of the technique to functionalize large-scale surfaces with nanowires at much lower cost than most of the existing techniques, i.e., a scalable technique.

Using the proposed technique, we have further examined polymerization in the confined nano-pores. Using PDMS nanowire in an AAO template with 75 nm pore-diameter and 1 um thickness, it was demonstrated that the cross-linking processes are significantly lengthened in the confined nanopores. Further experiments using materials with different monomer sizes reveal that the conformity of monomers plays a significant role in polymerization within a nano-confined space. Further research is needed in understanding this phenomenon.

## Figures and Tables

**Figure 1 micromachines-11-00046-f001:**
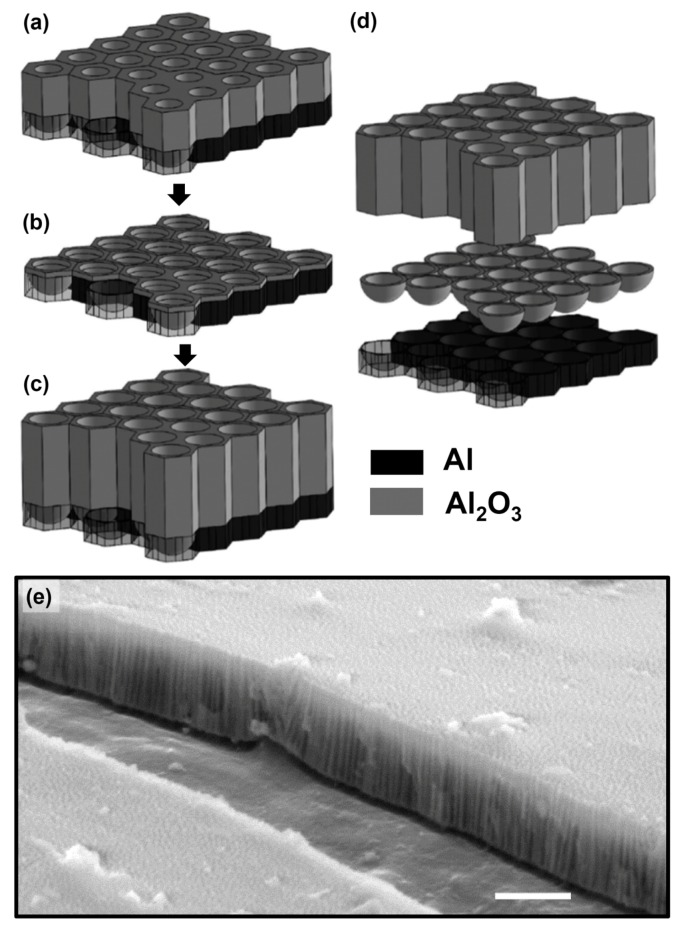
Fabrication processes of anodic aluminum oxide (AAO) template and the corresponding structures. (**a**) the 1st anodization to establish the electron density distribution over aluminum substrate; (**b**) etching procedure to create regularly allocated nano-bowls (i.e., barriers); (**c**) the second anodization at the same parameters as those used in (a) to create AAO nano-pores. (**d**) Sketch of sample AAO structures: from top to bottom are nano-tubes of alumina ($Al_{2}O_{3})$, alumina nano-bowls (barrier layer), and aluminum substrate. Color-coded by materials: (black) aluminum and (gray) alumina. (**e**) SEM micrograph of an AAO template anodized at 40 V at 1 °C in 0.3 M oxalic acid. The template was cleaned afterward with a reverse anodization process at −2 V at 20 °C in 0.2 M KCl solution for 150 s. Scale: 1 μm.

**Figure 2 micromachines-11-00046-f002:**
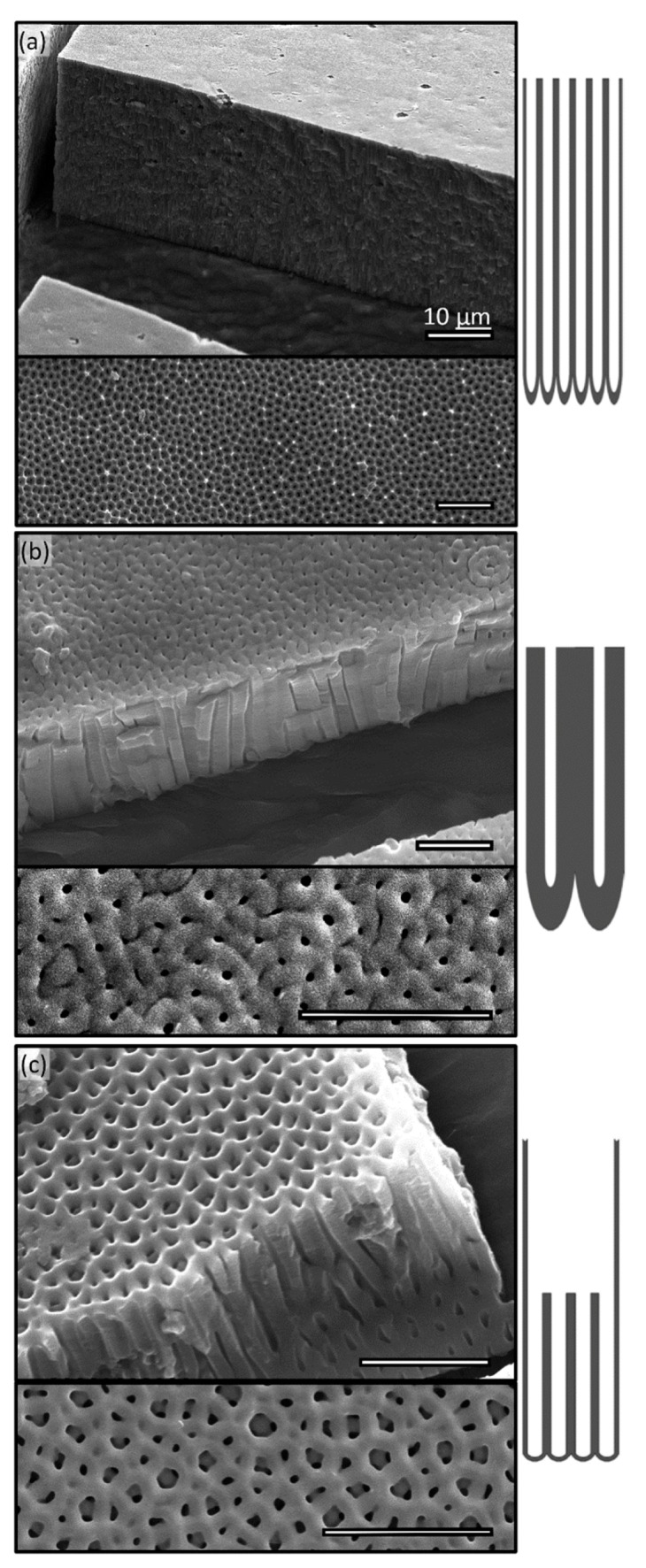
Topological features of AAO samples using different fabrication parameters: each consists of SEM micrographs (top) tilted at 15°, (bottom) from top view, and (side) illustration of AAO cross-section. Sample topological features include: (**a**) thin wall nano-pores with large aspect ratio (e.g., pore depth: diameter >100:1), which is anodized at 1 °C and 60 V for 4 h. (**b**) thick wall pores with a small pore diameter (e.g., ~50 nm), which is anodized at −3 °C and 140 V for 2 mins. (**c**) Tiered nano-pores with a layer of pores with larger pore-to-pore (p-p) distance on the top of a layer of pores with smaller p-p distance. The tiered pore structure is anodized at 1 °C and 40 V for 5 min, followed by anodization at 0 °C and 130 V, over an aluminum substrate conditioned by a first anodization and etching cycle at 1 °C and 40 V. Scale: 1 μm or otherwise stated.

**Figure 3 micromachines-11-00046-f003:**
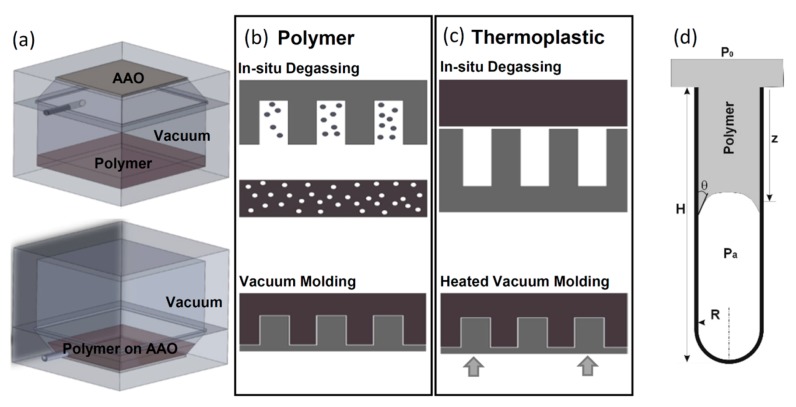
Polymeric nano-wire fabrication technique using vacuum-assisted molding into an AAO template. (**a**) Apparatus for performing in-situ vacuum-assisted molding. (**top**) Chamber, polymer, and template during the preparation phase, when they are vacuumed. (**bottom**) Chamber, polymer and template during the molding phase. (**b**) Vacuum-assisted molding process for polymers containing monomer and cross-linker includes in-situ degassing of template and polymer mixer (top) and vacuum molding (bottom). (**c**) Vacuum-assisted molding process for thermal plastics includes in-situ degassing of template and plastics (top) and vacuum molding with heat at an interfacial temperature exceeding glass transition temperature. (**d**) Schematics demonstrating the infiltration process of polymeric and/or plastic material into a nano-pore.

**Figure 4 micromachines-11-00046-f004:**
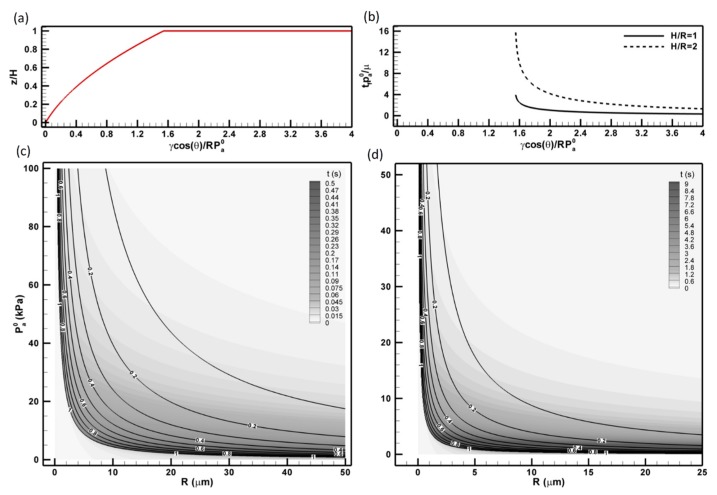
Modeling and characteristics of polymeric and plastic materials infiltrating a nano-pore. (**a**) normalized infiltration depth (z/H) *vs* the normalized capillary stresses ($\gamma \cos\theta /Rp_{a}^{0}$), (**b**) infiltration time ($t_{f}$) normalized by that of pressure-driven infiltration ($\mu /p_{a}^{0}$) *vs*
$\gamma \cos\theta /Rp_{a}^{0}$. Fabrication parameters, z/H (line contours) and $t_{f}p_{a}^{0}/\mu$ (flood contours) for (**c**) polydimethylsiloxane (cross-linking polymer) and (**d**) polytetrafluoroethylene (thermoplastics) with respect to initial pressure, $p_{a}^{0}$, and pore size, R. Properties of polydimethylsiloxane (PDMS) and polytetrafluoroethylene (PTFE) are listed in Table A2.

**Figure 5 micromachines-11-00046-f005:**
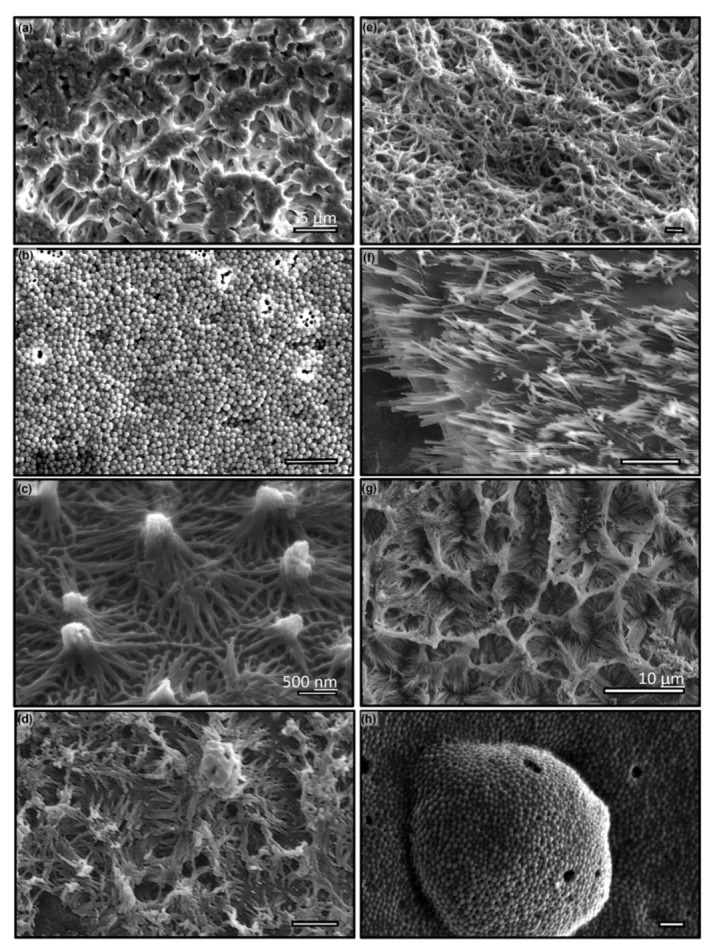
Sample of nanowires fabricated using vacuum molding over AAO template by various cross-linking polymers, (**a**) polydimethylsiloxane, (**b**) polyurethane (TC-9445) (**c**) epoxy, and (**d**) cyanoacrylate (crazy glue) and thermoplastic materials, (**e**) polyvinylidene fluoride, (**f**) polyamide, (**g**) polytetrafluoroethylene and (**h**) high-density polyethylene. The AAO templates used are 75 nm (a, c–d, g–h), 125 nm (b), and 55 nm (f). Scale: 1 µm or otherwise stated.

**Figure 6 micromachines-11-00046-f006:**
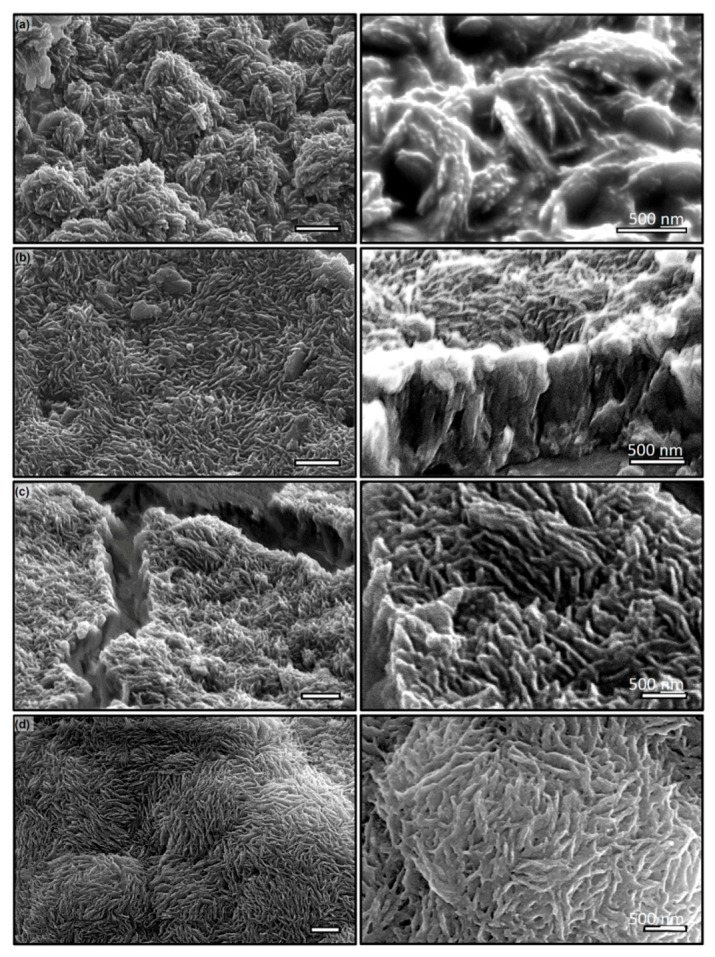
SEM micrographs of the PDMS nanowires over a 75 nm pore size and 2 μm thick AAO template cured for (**a**) 2 (**b**) 3 (**c**) 4 (**d**) 8 weeks, showing confinement effects. (**right**) Corresponding close-ups. Scale: 1 μm or otherwise stated.

**Figure 7 micromachines-11-00046-f007:**
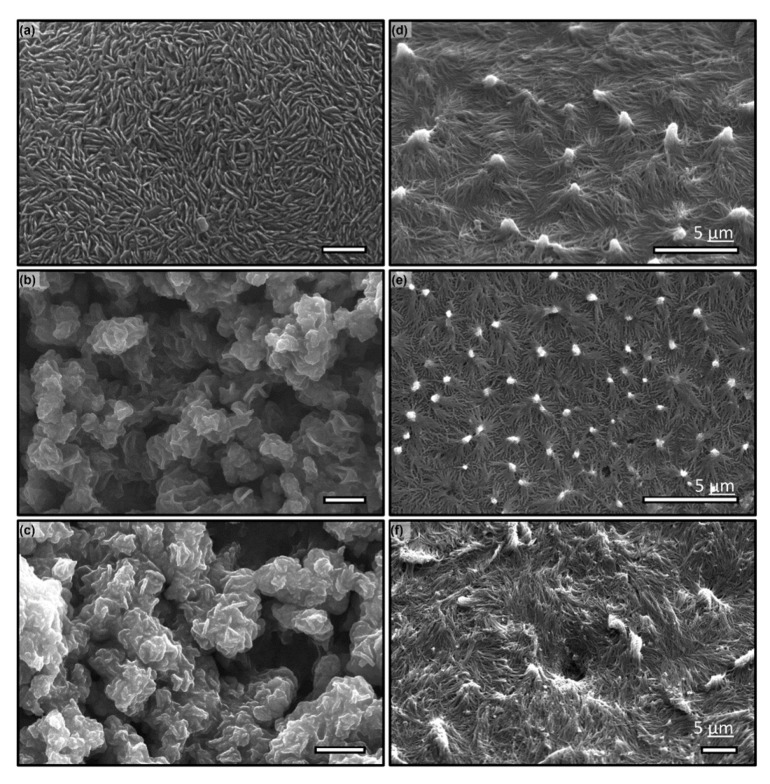
SEM micrographs of (**a**–**c**) PDMS and (**d**–**f**) epoxy using AAO templates of 150 nm, aspect ratio (AR) = 12 (**a**,**d**), 75 nm, AR = 16 (**b**,**e**), and 125 nm, AR = 40 (**c**,**f**).

**Table 1 micromachines-11-00046-t001:** Summary of materials and processing parameters used in fabricating nanowires.

Materials	Fabrication Parameters					Release Parameters		
Polymer	Molding			Curing		Etchant	T (°C)	Time (h)
	Vacuum (Pa)	Degas time(min)	Fill time (min)	T (°C)	Time (Day)			
Polydimethylsiloxane (PDMS)	2.7	>60	120	55	>27	1M NaOH	10	24–36
Polyurethane (TC9445)	2.7	12	5	55	7–10	1M NaOH	10	24–36
Polyurethane (V60)	2.7	12	5	55	7–10	1M NaOH	10	24–36
Epoxy (E60NC)	2.7	30	30	55	1–2	5M HCL	10	<24
Cyanoacrylate	2.7	20	5	20	<1	1M NaOH	10	<24
**Thermoplastics**	**Molding**					**Etchant**	**T (°C)**	**Time (h)**
	**Vacuum (Pa)**	**Degas time(min)**	**T (°C)**	**Mold Time (min)**				
Polyamide	2.7	15	220	5		5M HCL	6	<24
Polyvinylidene fluo-ride (PVDF)	2.7	15	180	5		5M HCL	6	<24
Polytetrafluoroethylene (PTFE)	2.7	15	320	5		5M HCL	6	<24
High Density Polyethylene (HDPE)	2.7	15	280	5		5M HCL	6	<24

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
