# Peer review of "Robust Fabrication of Polymeric Nanowire with Anodic Aluminum Oxide Templates"

_micromachines, 2019, doi:10.3390/mi11010046_

Round 1

Author Response

See attache file

Reviewer 2 Report

n this paper authors described a method to fabricate a polymeric nanowire by using  anodic aluminum oxide templates. The presented study could be interesting, however in the present form it is hard to read. Some part are extremely mixed and there are several lacks that it must be addressed in order to consider this manuscript for it publication.

I have added some comments (included, but not limited to) for authors:

In the introduction the lack of bibliographic references reduce its usage, the state of art is not adequately addressed in this section.

Materials and methods.

The materials are not present as a section, some of them are description of the methods some of them are correctly described and some are missing. Please provide the complete description of the materials, reagents, polymers….

Overall the experimental part is a mixture of the state of art, experimental description and results are mixed in this section, being this section extremely confusing. The manuscript must be re-written, and every part must be placed in the correct place. Include the Figures in the text. It is not clear why the temperatures are chosen, please provide the measurements of the glass transitions, and melting points of each polymer used.

Avoid the use of “we” form, and “a.k.a.” in the text. The correct abbreviation is Eq. and not Eqn.

Reviewer 3 Report

The paper is well organized and presented, the morphological results are well discussed. In my opinion, the paper will interest a high number of readers and can be published after control for some minor typos present in the text. See for example page 2 and figure 1 "electoral density distribution".

Round 2

Reviewer 2 Report

Authors have adequately improved their manuscript.